# Analysis of the incidence and influencing factors of high-risk foot in elderly patients with type 2 diabetes in a community in Beijing

**Gaoqiang Li[1‡], Qian Lu[2], Bing Wen[3], Huijuan Li[3], Jin Liu[3], Yanming Ding[4*]**

**1** Respiratory and Critical Care Medicine Department, Second Medical Center of Chinese People's Liberation Army General Hospital, Beijing, China, **2** Department of Surgical Nursing, School of Nursing, Peking University Health Science Center, Beijing, China, **3** Plastic Surgery and Burn Department, Peking University First Hospital, Beijing, China, **4** Chinese Nursing Association, Beijing, China

‡ GL as a first author.
* yanming_ding@126.com

## Abstract

### Background

Diabetic foot is one of the important causes of disability and death in diabetic patients, and effective measures can reduce or prevent the occurrence of diabetic foot, among which, the screening of diabetic high-risk foot helps to identify the risk groups that may progress to diabetic foot, and targeted intervention for this group of people can effectively reduce the prevalence of diabetic foot and the incidence of adverse consequences.

### Objective

To investigate the occurrence of high-risk foot with type 2 diabetes in elderly people in a community in Beijing and analyze its influencing factors, so as to provide evidence for preventing the occurrence of diabetic foot.

### Methods

269 elderly patients with type 2 diabetes in Xinjiekou community of Beijing were selected for foot examination by convenient sampling, and the high-risk foot was classified according to the international Diabetic Foot Working Group grading system.

### Results

The detection rate of high risk foot of type 2 diabetes was 51.7%. Binary Logistic regression analysis showed that age, glycosylated hemoglobin, hyperlipidemia, insulin therapy, and diabetic retinopathy were independent influencing factors for the occurrence of diabetic High-risk foot.

**Data availability statement:** All relevant data are within the manuscript and its Supporting Information files.

**Funding:** The author(s) received no specific funding for this work.

**Competing interests:** The authors have declared that no competing interests exist.

## Conclusion

The incidence of high-risk foot in elderly type 2 diabetes patients is high in this community. Community health care workers should pay attention to elderly diabetic patients with hyperlipidemia, insulin therapy, high glycated hemoglobin, and diabetic retinopathy to reduce the occurrence of foot ulcers.

## 1. Introduction

The risk of diabetes foot (DF) in diabetes patients' lifetime is 19% ~ 34%, and the risk of recurrence in 1-5 years after cure is as high as 40% ~ 65%, which has caused huge economic burden to families and society [1]. Especially in the elderly population, according to statistics, the prevalence of diabetes in the elderly population in China is 30.2% [2]. Diabetic foot has gradually become one of the most serious chronic complications affecting the quality of life of elderly diabetic patients. The guideline emphasizes the idea that prevention is more important than treatment for diabetic foot, and points out that prevention of DF is the key to reducing its incidence, amputation rate and mortality [3]. High-risk foot, as the early stage of DF, are defined as the presence of peripheral neuropathy and/or peripheral vascular disease and/or foot deformity, or a history of foot ulcers/amputations, but no active foot ulcers at present [4]. Some studies have pointed out that high risk foot patients can be screened through foot examination, and effective preventive measures can be taken to prevent 50% of patients from diabetes foot or amputation [5]. At present, about 90% of the elderly in China mainly provide for the aged at home, and most of the elderly do not pay enough attention to diabetes foot [6]. The key to prevent diabetes foot is whether diabetes patients have been properly assessed and intervened in the community [7]. Therefore, this study took elderly type 2 diabetes patients in the community as the research object, identified high-risk foot groups through foot examination, and analyzed its related factors, so as to provide reference for the prevention and management of chronic disease complications in the community.

## 2. Materials and methods

### 2.1. Ethics

We have obtained the approval of the Biomedical Research Ethics Committee of Peking University First Hospital, Ethics review number: (2018) Research No. (217). The research team obtained the informed consent of all participants in conducting this study, signed a written informed consent form, and complied with all regulations.

### 2.2. Study design and data source

This study is a cross-sectional design. A total of 269 elderly patients with type 2 diabetes in Xinjiekou community of Beijing were selected for questionnaire survey and foot examination from June 2019 to May 2020 by convenient sampling. Inclusion criteria (1) Age ≥ 60 years old. (2) Meet the diabetes diagnostic criteria established by WHO and are diagnosed with type 2 diabetes [8]. (3)No cognitive dysfunction and language communication disorders, can fill in the questionnaire independently or with the assistance of researchers. (4)life can be self-care, activities barrier-free, can come to the community hospital. (5)Volunteer to participate in this study and sign the informed consent. Exclusion criteria (1)Active diabetic foot ulcer. (2) gestational diabetes mellitus. (3)Secondary diabetes.

## 2.3. Methods

**2.3.1. Data collection and measurement.** Data collection includes two parts. The first part is self-designed questionnaire. The questionnaire included the patients' socio-demographic data (age, sex, etc.), physical examination (including height, weight, body mass index (BMI), blood pressure, waist circumference, hip circumference, waist-to-hip ratio), diabetes status (disease course, blood glucose treatment), complications (diabetic nephropathy, diabetic retinopathy), medical history (smoking history, hypertension, hyperlipidemia, hyperuricemia, coronary heart disease history, cerebrovascular history, cardiovascular surgery history, foot ulcer history, amputation history). The laboratory examination indexes (HbA1c, TC, TG, LDL, HDL) of the subjects were extracted from the medical records of community hospitals. The second part is foot examination, in which rigorously trained researchers examine the patient's peripheral arterial disease (intermittent claudication, resting pain, dorsalis pedis artery fluctuation, posterior tibial artery fluctuation, ABI measurement), peripheral neuropathy examination (10g nylon wire, 128Hz tuning fork, needle sensation, temperature sensation, Achilles tendon reflex, pain, numbness, abnormal sensation, etc.), and foot deformity examination (eversion, claw toe, hammer toe, Charcot foot, etc.).

**2.3.2. Diagnostic criteria for diabetic peripheral neuropathy (DPN).** According to the Guidelines for the Prevention and Treatment of Type 2 diabetes in China [9] (2017 version), it is suggested that the diagnostic criteria for DPN: for patients with clinical symptoms (pain, numbness, sensory abnormalities, etc.), any one of the five examinations (ankle reflex, acupuncture pain, vibration, pressure, temperature) is abnormal; Or for those without clinical symptoms, any 2 out of 5 tests are abnormal.

**2.3.3. Diagnostic criteria for peripheral arterial disease (PAD).** According to the Guidelines for the Prevention and Treatment of Type 2 diabetes in China [9] (2017 version), it is suggested that the diagnostic criteria for peripheral arterial disease (PAD) are Ankle-brachial index (ABI) ≤ 0.9, the pulse of the dorsum pedis or posterior tibial artery disappears, and there are symptoms of intermittent claudication or rest pain. Any one of the three abnormalities can be judged as peripheral arterial disease.

**2.3.4. Diagnostic criteria for foot deformity.** According to the Guidelines for the Prevention and Treatment of Type 2 Diabetes in China [9] (2017 edition), one of the symptoms, such as bunions, claw toes, hammer toes, and Charcot's feet, can be diagnosed as foot deformity.

**2.3.5. High-risk foot judgment.** In this study, the high-risk foot rating system of the International Diabetic Foot Working Group (IWGDF) was used as the criteria for evaluating High-risk foot [4]: Grade 0, that is, low risk feet, refers to those without DPN and PAD; Grade 1 high-risk foot refers to DPN only; Grade 2 high risk foot refers to DPN and foot deformity or PAD; Grade 3 high risk foot refers to those with a history of foot ulcers or amputations.

## 2.4. Statistical analysis

SPSS 20.0 statistical software was used for data analysis, with count data described as [n (%)], and chi square test was used for univariate analysis. Normal distribution metric data is expressed as mean ± standard deviation, and t-test is used for comparison between two groups. Non normal distribution metric data is expressed as median (interquartile range), and non parametric Mann Whitney rank sum test is used for comparison between two groups. Multivariate analysis is performed using binary logistic regression, and statistical significance is considered when $P < 0.05$.

## 3. Result

### 3.1. Sociodemographic characteristics and detection of high-risk foot of diabetes

Among the 269 patients, 129 were males and 140 were females. The mean age was $71.41 \pm 7.11$ years (60-94 years). The mean duration of diabetes was $14.72 \pm 8.62$ years, and the mean HbA1c was $(7.18 \pm 1.25)$%. According to the International Working Group on the Diabetic foot (IWGDF) high-risk foot classification system, the detection rate of type 2 diabetic high-risk foot in the elderly was 51.7%, of which 14.1% were grade 1 High-risk foot, 35.7% were grade 2 High-risk foot, and 1.9% were grade 3 High-risk foot.

### 3.2. Results of single factor analysis of influencing factors of high risk foot of elderly diabetes mellitus

By comparing the general data and disease related data of the elderly diabetes high-risk foot group and the elderly diabetes low-risk foot group, it is found by t-test or chi square test that there are statistically significant differences between the two groups in terms of age, diabetes course, glycosylated hemoglobin (HbA1c), blood glucose treatment methods (insulin injection group and non insulin injection group), diabetes retinopathy, hyperlipidemia, cerebrovascular surgery history, cardiovascular surgery history, etc. ($P < 0.05$), see Table 1 for details.

### 3.3. The results of binary logistic regression analysis on the influencing factors of high-risk foot of elderly diabetes

Taking whether there is a high-risk foot as the dependent variables and the statistically significant variables in the univariate analysis as the independent variable, binary logistic regression analysis was carried out. The results showed that age (OR = 1.083, 95% CI: $1.040 \sim 1.127$), HbA1c (OR = 1.269, 95% CI: $1.003 \sim 1.604$), diabetic retinopathy (OR = 2.240, 95% CI: $1.080 \sim 4.646$), hyperlipidemia (OR = 2.304, 95% CI: $1.248 \sim 4.255$), and the way of blood glucose treatment (OR = 2.597, 95% CI: $1.400 \sim 4.815$) were independent of the high-risk foot of elderly diabetes ($P < 0.05$), see Table 2 for details. The Hosmmer Lemeshow fitting test results show that the chi square value is 11.158, with a P-value of 0.193, which is greater than 0.05.

## 4. Discussion

As one of the serious chronic complications of diabetes, diabetic foot is characterized by high incidence, high disability rate and high fatality rate [3], and its mortality rate is much higher than that of most cancers [10]. Following the concept that prevention is more important than treatment, the screening of high-risk foot for diabetic patients and targeted intervention for high-risk foot patients can effectively reduce the occurrence of diabetic foot [11]. According to the International diabetes Foot Working Group (IWGDF) high risk foot grading system, the detection rate of high risk feet (Grade 1, Grade 2 and Grade 3) in this study is 51.7%. Compared with the results of Sun Shujuan et al. [12] (42.97%) and Banik et al. [13] (44.5%), the detection rate of high risk feet in this study is higher than both, which may be related to the relatively high age of the elderly in the community as the object of this study. The P value of Hosmmer Lemeshow fitting test is 0.193, which is greater than 0.05. The null hypothesis is accepted, indicating a good fit between the observed data and the regression model. The results analyzed by the binary logistic regression model truly and reliably reflect the true relationship between the original variables. Logistic regression analysis of this study showed that age, high glycosylated hemoglobin, retinopathy with diabetes, hyperlipidemia, and blood

**Table 1. Univariate analysis of risk factors for high-risk foot in an elderly community (n = 269).**

| Items | Low-risk foot group (n = 130) | High-risk foot group (n = 139) | Z/χ² | P |
|---|---|---|---|---|
| | M (Q₁ ~ Q₃)/n (%) | M (Q₁ ~ Q₃)/n (%) | | |
| **Age (years)** | 69.50 (66.00,74.25) | 71.00 (67.00,79.00) | -2.536 | 0.011* |
| **Duration of diabetes (years)** | 11.00 (6.75,16.00) | 17.00 (10.00,21.00) | -4.209 | <0.001* |
| **BMI (kg/m²)** | 24.70 (22.90,26.60) | 24.8 (22.80,27.30) | -0.270 | 0.787 |
| **HbA1c (%)** | 6.70 (6.30,7.40) | 7.00 (6.50,7.90) | -2.608 | 0.009* |
| **Waist circumference (cm)** | 86.00 (80.00,92.00) | 87.00 (80.00,93.00) | -0.212 | 0.832 |
| **Hip circumference (cm)** | 98.00 (93.00,103.00) | 99.00 (90.00,104.00) | -0.274 | 0.784 |
| **Waist-to-hip ratio** | 0.88 (0.86,0.90) | 0.88 (0.84,0.91) | -0.496 | 0.620 |
| **TC (mmol/L)** | 4.15 (3.66,4.87) | 4.23 (3.49,4.76) | -0.520 | 0.603 |
| **TG (mmol/L)** | 1.31 (0.99,1.84) | 1.24 (0.89,1.73) | -0.852 | 0.394 |
| **LDL-C (mmol/L)** | 2.07 (1.76,2.56) | 2.01 (1.62,2.48) | -1.150 | 0.250 |
| **HDL-C (mmol/L)** | 1.33 (1.17,1.52) | 1.29 (1.14,1.52) | -1.041 | 0.298 |
| **Gender** | | | | |
| Male | 57 (43.8) | 72 (51.8) | 1.702 | 0.192 |
| Female | 73 (56.2) | 67 (48.2) | | |
| **Blood glucose treatment** | | | | |
| Insulin injection group | 26 (20.0) | 60 (43.2) | 16.575 | <0.001* |
| Non-insulin injection group | 104 (80.0) | 79 (56.8) | | |
| **Diabetic Nephropathy** | | | | |
| Yes | 6 (4.6) | 13 (9.4) | 2.296 | 0.130 |
| No | 124 (95.4) | 126 (90.6) | | |
| **Diabetic retinopathy** | | | | |
| Yes | 14 (10.8) | 37 (26.6) | 10.983 | 0.001* |
| No | 116 (89.2) | 102 (73.4) | | |
| **Smoking history** | | | | |
| Yes | 24 (18.5) | 29 (20.9) | 0.245 | 0.621 |
| No | 106 (81.5) | 110 (79.1) | | |
| **Hypertension** | | | | |
| Yes | 105 (80.8) | 111 (79.9) | 0.035 | 0.851 |
| No | 25 (19.2) | 28 (20.1) | | |
| **Hyperlipemia** | | | | |
| Yes | 81 (62.3) | 108 (77.7) | 7.615 | 0.006* |
| No | 49 (37.7) | 31 (22.3) | | |
| **Hyperuricemia** | | | | |
| Yes | 12 (9.2) | 23 (16.5) | 3.177 | 0.075 |
| No | 118 (90.8) | 116 (83.5) | | |
| **History of coronary heart disease** | | | | |
| Yes | 41 (31.5) | 45 (32.4) | 0.022 | 0.883 |
| No | 89 (68.5) | 94 (67.6) | | |
| **History of cerebrovascular surgery** | | | | |
| Yes | 16 (12.3) | 30 (21.6) | 4.076 | 0.043* |
| No | 114 (87.7) | 109 (78.4) | | |

*(Continued)*

**Table 1.** (Continued)

| Items | Low-risk foot group (n = 130) | High-risk foot group (n = 139) | Z/χ² | P |
|---|---|---|---|---|
| | M (Q₁ ~ Q₃)/n (%) | M (Q₁ ~ Q₃)/n (%) | | |
| History of cardiovascular surgery | | | | |
| Yes | 15 (11.5) | 29 (20.9) | 4.269 | 0.039* |
| No | 115 (88.5) | 110 (79.1) | | |

**Note:** TC: Total cholesterol, TG: Triglyceride, LDL: Low-Density Lipoprotein Cholesterol, HDL: High-Density Lipoprotein Cholesterol.

**Table 2. Logistic regression analysis of risk factors for high-risk foot in an elderly community.**

| Items | B | SE | Wald | P | OR | 95%CI |
|---|---|---|---|---|---|---|
| Age | 0.080 | 0.020 | 15.143 | 0.000 | 1.083 | 1.040 ~ 1.127 |
| HbA1c (%) | 0.238 | 0.120 | 3.947 | 0.047 | 1.269 | 1.003 ~ 1.604 |
| Diabetic retinopathy | 0.806 | 0.372 | 4.693 | 0.030 | 2.240 | 1.080 ~ 4.646 |
| Hyperlipemia | 0.835 | 0.313 | 7.111 | 0.008 | 2.304 | 1.248 ~ 4.255 |
| Insulin injection group | 0.954 | 0.315 | 9.175 | 0.002 | 2.597 | 1.400 ~ 4.815 |
| Constant | -8.365 | 1.870 | 20.009 | 0.000 | 0.000 | |

$R^2 = 0.214.$

Note: B: regression coefficient, SE: standard Error, Wald: chi square value, P: p-value, OR: odds Ratio, 95%CI: 95%confidence Interval.

glucose treatment (insulin injection group) were independent risk factors for high-risk foot of elderly diabetes.

Age is the risk factor of high-risk foot in elderly patients with diabetes, which is consistent with the research results of Kisozi et al. [14]. In this study, the average age of the elderly people with high-risk foot of diabetes is 72.57 ± 7.65 years old, which is larger than the average age of the people with low-risk foot (70.17 ± 6.27 years old). Logistic regression analysis found that for every 1-year-old increase in age, the risk of developing high-risk foot increased by 1.083 times. The reason may be that with the increase of age, the nerve conduction delay is longer, the amplitude is smaller, and the conduction speed is slower [15].

The increase of HbA1c indicates that the blood sugar level of patients is poorly controlled. Long-term high blood sugar will cause non-enzymatic glycosylation of amino acid components on the blood vessel wall, resulting in vascular smooth muscle hyperplasia, aggravating the occurrence of atherosclerosis, and increasing the risk of peripheral vascular diseases [16,17]. At the same time, this study found that the risk of developing high-risk foot in insulin users was 2.597 times that of non-insulin users. It is generally believed that insulin users have a longer course of diabetes, which may be related to the long-term hyperglycemia causing swelling, degeneration and even necrosis of nerve fibers, leading to nerve demyelination, and then peripheral neuropathy [18].

Diabetic retinopathy is an independent risk factor for high-risk foot, which is consistent with the conclusions of Al-Rubeaan et Al. [19]. The reason may be that the two have some common pathophysiological basis, that is, multiple mechanisms such as oxidative stress mediated by long-term hyperglycemia, glycosylation end products and chronic inflammatory response [20], which ultimately lead to peripheral nerve and vascular lesions.

Dyslipidemia is an independent risk factor for high risk foot. Data from this study showed that patients with hyperlipidemia had a 2.304 times higher risk of developing High-risk foot than those without. Consistent with the conclusions of many studies [21–23], on the one

hand, it may be related to hyperlipidemia promoting the formation of atherosclerotic plaque; On the other hand, the reason may be related to the oxidative stress and endothelial dysfunction of dorsal root ganglion sensory neurons caused by abnormal blood lipids, which in turn leads to peripheral neuropathy [24].

Finally, it should be pointed out that due to the limitations of conditions, random sampling was not realized, but convenient sampling was chosen. Therefore, the generalizability of this study has some potential limitations.

## 5. Conclusion

Among the elderly in the community, the overall detection rate of high-risk foot of diabetes is high, and it is significantly related to the elderly, elevated glycosylated hemoglobin, hyperlipidemia, blood glucose treatment, and diabetes retinopathy. The research results show that community medical staff should regularly screen the feet of elderly diabetes patients in the community according to the recommendations of IWGDF, and take targeted measures according to the classification of High-risk foot to reduce the harm of diabetes feet to patients, families and society.

## Supporting information

**S1 Dataset. This is the S1 Dataset Title.**
(SAV)

## Acknowledgments

The authors would like to thank the staff of Xinjiekou Community Hospital for their strong support, and the participants who volunteered to participate in this study.

## Author contributions

**Conceptualization:** Qian Lu, Bing Wen, Yanming Ding.

**Data curation:** Gaoqiang Li.

**Formal analysis:** Gaoqiang Li, Qian Lu, Huijuan Li, Yanming Ding.

**Funding acquisition:** Bing Wen, Yanming Ding.

**Investigation:** Gaoqiang Li, Qian Lu, Huijuan Li, Jin Liu, Yanming Ding.

**Methodology:** Gaoqiang Li, Bing Wen, Huijuan Li, Jin Liu.

**Supervision:** Qian Lu, Bing Wen, Jin Liu, Yanming Ding.

**Writing – original draft:** Gaoqiang Li, Yanming Ding.

**Writing – review & editing:** Qian Lu.

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
