## [Decision Letter · Decision Letter 0]

10 Jan 2025

PONE-D-24-45712Analysis of the incidence and influencing factors of high-risk foot in elderly patients with type 2 diabetes in a community in BeijingPLOS ONE

Dear Dr. Ding,

Thank you for submitting your manuscript to PLOS ONE. After careful consideration, we feel that it has merit but does not fully meet PLOS ONE’s publication criteria as it currently stands. Therefore, we invite you to submit a revised version of the manuscript that addresses the points raised during the review process.

We look forward to receiving your revised manuscript.

Kind regards,

Sanaullah Sajid, M.Phil/PhD

Academic Editor

PLOS ONE

**Journal Requirements:**

Reviewers' comments:

Reviewer's Responses to Questions

**Comments to the Author**

1. Is the manuscript technically sound, and do the data support the conclusions?

Reviewer #1: Yes

Reviewer #2: Yes

2. Has the statistical analysis been performed appropriately and rigorously? 

Reviewer #1: Yes

Reviewer #2: Yes

3. Have the authors made all data underlying the findings in their manuscript fully available?

Reviewer #1: No

Reviewer #2: Yes

4. Is the manuscript presented in an intelligible fashion and written in standard English?

Reviewer #1: Yes

Reviewer #2: Yes

5. Review Comments to the Author

**Reviewer #1:**  1. The authors collected and measured data related with diabetic foot risk for 269 elderly patients with type 2 diabetes in a Chinese community. They detected the incidence rate of high-risk foot and identified influencing factors using univariate hypothesis tests and multivariate logistic regression. The results highlighted the importance for examination of high-risk foot in elderly diabetic patients, and proposed several risk factors that warrant attention.

2. In the univariate analysis part, t test requires data to be normal distribution, so it would be more rigorous to test the normality of data distribution before t test, or to use the nonparametric Wilcoxon rank sum test would be safe. After performing hypothesis tests for each factor in Table 1, it would be more rigorous to perform FDR (false discovery rate)-controlling procedure to ensure the overall false positivity lower than a given level. In the logistic regression part, the accuracy of coefficient estimates and the calculation of p-values often rely on some assumptions for the data, so it would be more rigorous to perform some regression diagnostics for the data and results, if convenient.

3. The original data from the questionnaire, medical records, and foot examination are not available in the manuscript or supporting information. If without special restrictions, the original data should be provided.

4. The abbreviations in the tables should be better explained in the footnotes, such as TC, TG, LDL-C, HDL-C in Table1; B, SE, Wald, OR, CI in Table2. The table titles should be "risk factors for high-risk foot" rather than "risk factors for type 2 diabetes", and "in an elderly community" rather than "in elderly communities". The significant p-values in the tables should be better marked, for example, using asterisks.

Some typos should be checked. (1) Whether to capitalize the first letters, such as headings and subheadings, the table items, etc. The first word after semicolon should not be capitalized. (2) Brackets appear to be typed in both Chinese and English modes. (3) Space should be inserted between one word and its successive left bracket, between one word and its front comma or period. (4) "High-risk foot" or "High-risk feet", there are two different expressions across the manuscript. (5) Singular or plural nouns, such as "statistically significant variables" and "independent variables" in Section 3.3. (6) "HBA1c" should be "HbA1c" in Section 4.

**Reviewer #2:**  Minor concerns:

Since convenient sampling was used, the authors may consider acknowledging the potential limitation regarding generalizability.

The manuscript could be further improved by discussing possible confounding factors that might influence the outcomes.

6. PLOS authors have the option to publish the peer review history of their article (what does this mean? ). If published, this will include your full peer review and any attached files.

**Do you want your identity to be public for this peer review?** For information about this choice, including consent withdrawal, please see our Privacy Policy .

Reviewer #1: No

Reviewer #2: **Yes: ** Dr. Samiullah Sajid

---

## [Author Response · Author response to Decision Letter 1]

28 Jan 2025

Dear reviewers:

Thank you very much for your review and valuable suggestions. I have made comprehensive and careful revisions to the article based on your valuable feedback. The specific revision suggestions are as follows:

Reviewer #1: 

1. The authors collected and measured data related with diabetic foot risk for 269 elderly patients with type 2 diabetes in a Chinese community. They detected the incidence rate of high-risk foot and identified influencing factors using univariate hypothesis tests and multivariate logistic regression. The results highlighted the importance for examination of high-risk foot in elderly diabetic patients, and proposed several risk factors that warrant attention.

Author: Dear reviewer, thank you very much for reviewing and affirming my manuscript. I will carefully revise it based on your valuable suggestions.

2.In the univariate analysis part, t test requires data to be normal distribution, so it would be more rigorous to test the normality of data distribution before t test, or to use the nonparametric Wilcoxon rank sum test would be safe. After performing hypothesis tests for each factor in Table 1, it would be more rigorous to perform FDR (false discovery rate)-controlling procedure to ensure the overall false positivity lower than a given level. In the logistic regression part, the accuracy of coefficient estimates and the calculation of p-values often rely on some assumptions for the data, so it would be more rigorous to perform some regression diagnostics for the data and results, if convenient.

Author: Dear reviewer, based on your valuable suggestions, I have carefully verified the normality of the data and found that the metric data does not meet the normality criteria. Therefore, following your suggestion, I have used a non parametric Mann Whitney rank sum test with two independent samples. In the regression analysis, I conducted a Hosmmer-Lemeshow fit test, and the chi square value of the test results was 11.158, with a P value of 0.193, which is greater than 0.05. Accepting the null hypothesis, it indicates that the observed data fits well with the regression model, and the results obtained from the binary regression model truly and reliably reflect the true relationship between the original variables.The above sections have been supplemented in the Results and Discussion sections.

3.The original data from the questionnaire, medical records, and foot examination are not available in the manuscript or supporting information. If without special restrictions, the original data should be provided.

Dear reviewer, I have uploaded the relevant raw data in the form of attachments to the submission system as per your request.

4.The abbreviations in the tables should be better explained in the footnotes, such as TC, TG, LDL-C, HDL-C in Table1; B, SE, Wald, OR, CI in Table2. The table titles should be "risk factors for high-risk foot" rather than "risk factors for type 2 diabetes", and "in an elderly community" rather than "in elderly communities". The significant p-values in the tables should be better marked, for example, using asterisks.

Author: Dear reviewer, thank you for your valuable suggestions. I have made revisions in all corresponding areas based on your suggestions. Please refer to the "Revised Manual with Track Changes" document for specific modifications.

Some typos should be checked. (1) Whether to capitalize the first letters, such as headings and subheadings, the table items, etc. The first word after semicolon should not be capitalized. (2) Brackets appear to be typed in both Chinese and English modes. (3) Space should be inserted between one word and its successive left bracket, between one word and its front comma or period. (4) "High-risk foot" or "High-risk feet", there are two different expressions across the manuscript. (5) Singular or plural nouns, such as "statistically significant variables" and "independent variables" in Section 3.3. (6) "HBA1c" should be "HbA1c" in Section 4.

Author: Dear reviewer, I have made revisions and improvements to each item based on your valuable suggestions. Please refer to the "Revised Manuscript with Track Changes" document for specific modifications.

Reviewer #2: Minor concerns:

Since convenient sampling was used, the authors may consider acknowledging the potential limitation regarding generalizability.

The manuscript could be further improved by discussing possible confounding factors that might influence the outcomes.

Author: Dear reviewer, thank you very much for your valuable suggestion. Based on your suggestion, I have added it to the discussion section of the article, specifically "Finally, it should be pointed out that due to the limitations of conditions, random sampling was not realized, but conventional sampling was chosen. There before, the generality of this study has some potential limitations".

---

## [Editor Report · Decision Letter 1]

5 Feb 2025

Analysis of the incidence and influencing factors of high-risk foot in elderly patients with type 2 diabetes in a community in Beijing

PONE-D-24-45712R1

Dear Dr. Ding,

We’re pleased to inform you that your manuscript has been judged scientifically suitable for publication and will be formally accepted for publication once it meets all outstanding technical requirements.

Kind regards,

Sanaullah Sajid, M.Phil/PhD

Academic Editor

PLOS ONE
---

## [Editor Report · Acceptance letter]

PONE-D-24-45712R1

PLOS ONE

Dear Dr. Ding,

I'm pleased to inform you that your manuscript has been deemed suitable for publication in PLOS ONE. Congratulations! Your manuscript is now being handed over to our production team.

Kind regards,

on behalf of

Dr. Sanaullah Sajid

Academic Editor

PLOS ONE